# Evaluating Soil–Root Interaction of Hybrid Larch Seedlings Planted under Soil Compaction and Nitrogen Loading

**Tetsuto Sugai** [1] , **Satoko Yokoyama** [2] , **Yutaka Tamai** [3] , **Hirotaka Mori** [3] , **Enrico Marchi** [4] ,
**Toshihiro Watanabe** [1] , **Fuyuki Satoh** [5] **and Takayoshi Koike** [1,2,*]

1   Plant Nutrition laboratory, Hokkaido University, Sapporo 060-8689, Japan;
    tsugai@for.agr.hokudai.ac.jp (T.S.); nabe@chem.agr.hokudai.ac.jp (T.W.)
2   Silviculture and Forest Ecological Studies, Hokkaido University, Sapporo 060-8589, Japan;
    s-yokoyama@frontier.hokudai.ac.jp
3   Forest Bioresource Technology laboratory, Hokkaido University, Sapporo 060-8589, Hokkaido, Japan;
    ytamai@for.agr.hokudai.ac.jp (Y.T.); mrh28m@eis.hokudai.ac.jp (H.M.)
4   Department of Agriculture, Food, environment and Forestry, University of Florence, Via S. Bonaventura 13,
    50145 Firenze, Italy; enrico.marchi@unifi.it
5   Field Science Center for Northern Biosphere, Hokkaido University, Sapporo 060-8589, Hokkaido, Japan;
    f-satoh@fsc.hokudai.ac.jp
*   Correspondence: tkoike@for.agr.hokudai.ac.jp

**Abstract:** Although compacted soil can be recovered through root development of planted seedlings, the relationship between root morphologies and soil physical properties remain unclear. We investigated the impacts of soil compaction on planted hybrid larch $F_1$ (*Larix gmelinii* var. *japonica* × *L. kaempferi*, hereafter $F_1$) seedlings with/without N loading. We assumed that N loading might increase the fine root proportion of $F_1$ seedlings under soil compaction, resulting in less effects of root development on soil recovery. We established experimental site with different levels of soil compaction and N loading, where two-year-old $F_1$ seedlings were planted. We used a hardness change index (HCI) to quantify a degree of soil hardness change at each depth. We evaluated root morphological responses to soil compaction and N loading, focusing on ectomycorrhizal symbiosis. High soil hardness reduced the total dry mass of $F_1$ seedlings by more than 30%. Significant positive correlations were found between HCI and root proportion, which indicated that $F_1$ seedling could enhance soil recovery via root development. The reduction of fine root density and its proportion due to soil compaction was observed, while these responses were contrasting under N loading. Nevertheless, the relationships between HCI and root proportion were not changed by N loading. The relative abundance of the larch-specific ectomycorrhizal fungi under soil compaction was increased by N loading. We concluded that the root development of $F_1$ seedling accelerates soil recovery, where N loading could induce root morphological changes under soil compaction, resulting in the persistent relationship between root development and soil recovery.

**Keywords:** soil compaction; N loading; fine root; root morphology; ectomycorrhizal fungi

## 1. Introduction

Impacts of forest soil compaction that are caused by forestry machine operation have serious concerns for sustainable forest management [1–3]. Soil compaction degrades forest ecosystem functions [4], such as nutrient availability [5,6], the community of soil macro- and microorganism [7], and tree growth [8–10]. High soil hardness can inhibit root development, resulting in the inhibition of

physiological activities and growth suppression of trees [4]. Soil hardness means pressure resistance and it is a general physical parameter [11–13]. The effects of soil compaction on tree growth vary depending on soil type, condition, and tree species [14–16]. Manipulation experiments can contribute to more accurate understanding of tree growth response to soil compaction and its mechanisms [17–19].

The recovery of soil structures and functions from compaction (hereafter, soil recovery) varies depending on several site-related factors, such as soil texture, organic matter content, pedoclimate, and biomass and activity of soil biota [20–23]. Soil recovery can be associated with the root growth of planted tree seedlings, which has been primarily studied in alder species (*Alnus* sp.) [24,25], although the alder planting has raised other ecological concerns in forest management [26]. Root development can penetrate the compacted soil layer [27] and form cracks and fissures in soil [28,29]. Root development under compacted soil would be affected by soil biotic and abiotic factors [30]. Although fertilizer application could promote both aboveground and belowground growth, even in compacted soil [31,32], few studies have elucidated root morphological responses and the symbiosis with soil microorganisms under soil compaction and N loading [33]. Because artificial construction to improve compacted soil properties has economic and ecological issues [6,24], it is of outmost importance to investigate the relationship between roots of planted seedlings and soil physical properties for the ecological managements of forest soil [1].

Larch (*Larix* sp.) trees have widely been planted for timber production in Northeast Asia [34,35]. Among larch species, hybrid larch $F_1$ (hereafter $F_1$) has been developed for the superior initial growth [34,36]. However, the overuse of forestry machines has been concerned in afforestation regions [37–39]. The effects of N loading on tree growth and the symbiosis between ectomycorrhizal (ECM) fungi have been extensively investigated [40,41]. N loading promotes the growth of $F_1$ seedling [42], whereas excess N loading would reduce its stress resistance [43]. A stable ECM community of $F_1$ seedlings was observed under N loading [41]. Therefore, we supposed that $F_1$ would be a suitable model tree species to verify the responses of root development to soil compaction and N loading under the consistent relationship with ECM fungi.

In this study, we investigated the growth of $F_1$ seedlings planted in the compacted soil where N was added. Given the root response to soil physical [31] and nutritional conditions [43], we hypothesized that N loading would increase the fine root proportion of $F_1$ seedlings under soil compaction, resulting in a reduced effect of root development on soil recovery. We attempted to verify the following objectives: (1) whether the root developments of $F_1$ would affect soil hardness and (2) how the morphological traits respond to soil compaction and N loading. We synthesized these results to reveal the capacity of promoting soil recovery by planting seedlings for forest ecological management [44].

## 2. Material and Methods

### 2.1. Site and Experimental Design

This experiment was conducted in Sapporo Experimental Forest of Hokkaido University, Japan (43°04′ N, 141°20′ E, 15 m a.s.l.), during two growing seasons from May 2018 to November 2019. The climate is cool temperate with high humidity. In 2018, the mean annual air temperature was 9.1 °C and the total precipitation was 1282 mm and, in 2019, the mean annual air temperature was 9.5 °C and the total precipitation was 814 mm. The snow-free period is mid-April to early November. Meteorological data were recorded by the Japan Meteorological Agency near the study site. The experimental plots were established at the nursery under full sunlight to make them as uniform as possible. The soil of the experimental area was classified as brown forest soil (Dystric Cambisols) [45].

We examined the effects of soil compaction and N loading using a split-plot design (Figure S1). Six combinations were set with five replicate plots for each treatment (three levels of soil compaction × 2 N loading levels × 5 replicated blocks). First, a 40 m track line for each compaction level was divided into five blocks. The distance from each track line was 1 m, with a trench of 40 cm depth. Each block size was approximately 1.5 m × 6.0 m, and each block was separated by at least 1 m.

Each block was divided into two subplots for N loading treatment (a total of 30 subplots). A subplot size was approximately set as 1.5 m × 2.5 m with a distance of 1 m from each other.

One-year-old seedlings of hybrid larch $F_1$ (*L. gmelinii* var. *japonica* × *L. kaempferi*) were produced at a nursery of Hokkaido Research Organization, Forestry Research Institute. Before needle unfolding, four seedlings were transplanted into all subplots on 15 May 2018 (a total of 120 seedlings). A seedling was planted into the space excavated (0.1 m × 0.1 m × 0.4 m), and the distance from each seedling was at least 0.5 m in a subplot. The initial size (means ± SD) was 25.53 ± 2.17 cm for aboveground height and 6.00 ± 0.87 mm for collar diameter. After planting, germinated grasses that were around the $F_1$ seedlings were manually removed.

A root tillage treatment was applied by a cultivator for mixing the soil to a depth of approximately 30 cm in order to set high environmental homogeneity of the study site. Weeding was carried out before the soil compaction treatment. The soil was compacted with heavy machine passes conducted on 11 May 2018. The target levels of surface soil hardness were set as 4.7 and 14.0 kg cm$^{-2}$ based on the national environmental standards [46]. The heavy machines passed over the study sites until the target of the soil compaction level was achieved, as measured using a soil hardness tester (Yamanaka's Soil Hardness tester, Fujiwara Scientific Co., Ltd., Tokyo, Japan). Two types of heavy machinery were operated on 1.5 m wide and 40 m long tracks: a 1.0 t mini excavator (U-008, KUBOTA, Osaka, Japan) was used for the low compaction level and a 2.0 t tractor (TA278F, ISEKI, Matsuyama, Japan) was used for the high compaction level. Three levels of soil compaction were simulated: control, undisturbed soil after tillage; low compaction treatment obtained with 10 passes of a mini excavator after tillage; and, high compaction obtained with 23 passes of a tractor after tillage.

Two levels of N loading were simulated: 0 (−N) and 50 kg N ha$^{-1}$ year$^{-1}$ (+N). The N loading level was reflected with the observed maximum value in Japan [47]. Ammonium sulfate $((NH_4)_2SO_4)$ was used because of concerns of the recent N deposition in Japan [48]. In plots of +N, N loading was conducted on 6 July, 20 July, 17 August, 25 September, and 18 October 2018, without rainy days. $(NH_4)_2SO_4$ was dissolved in tap water and then applied directly on soil, avoiding the application on seedling bodies. The applied volume for a seedling was approximately 500 mL during each application. Plots of −N were irrigated with the same amount of tap water on the treatment dates.

### 2.2. Soil Environment

In this study, soil hardness (SH) was defined as the pressure resistance acquired by a conical tip probe pushing into soil using a soil hardness tester (Yamanaka's Soil Hardness tester, Fujiwara Scientific Co., Ltd., Tokyo, Japan). Soil bulk density (SBD) is an indicator of soil packing degree and it is calculated as the dry weight per unit volume of soil.

The vertical SH was measured along with four depths: 0–10, 10–20, 20–30, and 30–40 cm. These measurements were taken during the first and final research periods. The first SH along with depth were measured on 15 May 2018. A mini pit was manually dug into soil with a volume of approximately 10 cm × 35 cm × 45 cm. The position of the hole was randomly set in each subplot at least 30 cm away from the planted seedling. Subsequently, SH was randomly measured in a soil face at each depth. The mean value of these measurements was used as a representative SH value for each seedling. The final SH along with depth was measured in a space with a volume of approximately 50 cm × 50 cm × 40 cm from the end of October to early November in 2019, where a seedling was excavated. The three points where the SH was measured were randomly selected at each depth after excavating a seedling, and the mean value was calculated for each seedling. The hardness change index (hereafter HCI) was calculated as the difference between the final and first mean values of SH to evaluate the relationship between soil variation and root development considering soil depth.

The surface SH was also measured on 15 May, 5 June, 20 July, 11 September, and 23 October 2018, and on 1 May, 3 September, and 2 November 2019. The surface SH was measured at four to six random points approximately 20 cm away from the seedling stem. The mean value of the points was used as a representative SH value for each subplot.

Surface soil sampling was conducted in order to measure SBD and soil water contents (SWC) on 5 June, 20 July, 11 September, and 23 October 2018, avoiding the rainy days. Soil was sampled at each subplot using a metal cylinder (5 cm × 20 cm$^2$, Daiki Rika Kogyo Co., Ltd., Saitama, Japan) without any plant litter. After sampling, the fresh soil mass was immediately weighed using a 0.1 g scale (EB-3300SW, Shimadzu Co., Ltd., Kyoto, Japan) and then dried for 24 h at 105 °C.

On 15 August 2018, another soil sampling was conducted at each subplot in order to determine three phase fractions of soil. Solid and liquid phases were measured using a digital volume analyzer (DIK-1150, Daiki Rika Kogyo Co., Ltd., Saitama, Japan), and the residual was calculated as the air phase fraction.

Fresh soil was sampled for its chemical properties on August 2018. The sampled soil was passed through a stainless-steel sieve with a 1.0 mm mesh. Subsequently, 10 g of soil was mixed with 25 mL of 2 M potassium chloride solution. The samples were shaken for over 1 h before measurement using a pH sensor (RM-30P, TOA DKK, Co., Ltd., Tokyo, Japan). The soil inorganic N contents of the same samples were measured. The samples were filtered through a 1-μm filter (No. 5C filter paper, Toyo Roshi, Tokyo, Japan) for the colorimetric measurement of inorganic N (total value of $NH_4^+$–N and $NO_3^-$–N) using a flow injection analyzer (AQLA-700, Aqualab Co., Ltd., Tokyo, Japan).

### 2.3. Seedling Performances

#### 2.3.1. Aboveground

The height and the collar diameter of 120 seedlings were regularly measured using a measuring tape with 0.1 cm resolution and a digital caliper with 0.01 mm resolution (Mitsutoyo, Kanagawa, Japan) on 10 June, 6 July, 7 August, 5 September, and 8 October 2018, and 14 October 2019. The diameter was calculated as the mean of two crosswise measurements. The xylem pressure potential of the shoot tips was measured on 9 September 2019. Five seedlings were randomly selected for each treatment, and the sampled intact shoot was measured using a pressure chamber (PMS-600, PMS Instrument Co., Albany, OR, USA) at predawn (03:30–04:30 h) and mid-day (13:00–14:00 h).

On 21 October 2019, the aboveground plant body of 120 seedlings was harvested. The harvested biomass of each seedling was divided by stems, branches, and needles. The stems were divided into three sections at the internode: a current-year stem grown in 2019, a one-year stem formed in 2018, and the two-year stem. The height and collar diameter of these stems were measured using the same methods described above. All of the branches with needles were divided into three sections, following the same approach used for the stems. The branches of each stem were counted. The longest branch was sampled in a current-year stem, and its length and bottom diameter were measured. Subsequently, all of the separated organs were put into a dry oven at 80 °C for two weeks to determine the dry mass. We defined the branch productivity index (BPI) and needle productivity index (NPI) as the following formulas in order to evaluate the morphological balance and potential productivity in a current-year stem:

$$BPI\left(\text{m}^{-1}\right) = \frac{The\ number\ of\ branch\ in\ current-year\ stem}{The\ length\ of\ current\ stem\ (\text{m})}, \tag{1}$$

$$NPI\left(\text{g m}^{-1}\right) = \frac{Needle\ dry\ weight\ in\ current-year\ stem\ (\text{g})}{The\ length\ of\ maximum\ current-year\ branch\ (\text{m})}. \tag{2}$$

#### 2.3.2. Belowground

The root systems of the seedlings were carefully excavated at the control and at a high level of soil compaction treatment from the end of October to early November in 2019. The root systems were sampled for three seedlings of each subplot (two compaction levels × 2 N loading levels × 5 replicated blocks × 3 biological replicates, a total of 60 seedlings). Before sampling, the excavating area and depth

were set to approximately 50 cm × 50 cm × 40 cm. The excavated volume of space was measured using a measuring tape with 1 cm resolution just after root sampling to avoid any soil deformation.

The root system was carefully washed in order to eliminate soil and separated visually into a rootstock and other roots. The other roots were divided into lateral roots (diameter ≥ 2 mm) and fine roots (diameter < 2 mm) using a digital caliper. Finally, the separated roots were dried in an oven at 65 °C for a week, and the dry weight was measured. These weights were used to calculate root density as the ratio between the root weight and excavated soil volume.

When excavating root systems of seedlings, a seedling was randomly selected in each subplot in order to determine the fine root morphological traits and the ECM association (two compaction levels × 2 N loading levels × 5 replicated blocks × 1 biological replicates, a total of 20 seedlings). Approximately six fine roots were collected along each distance from stem (0–10, 10–20, and 20–30 cm). The sampled roots were covered with wet papers, stored in a plastic bag, and transferred to a dark refrigerator (4 °C) in the laboratory until the following measurements. The remaining soil was carefully removed from fine roots using a brush. Three intact fine roots were randomly selected from each seedling and scanned using a double-lamp bed scanner with 800 dpi (GT-X 970, Epson, Japan). The scanned image was analyzed to determine the total length, surface area, and volume of each fine root while using root analysis software WinRHIZOTRON 2012 (Regent Instruments, Quebec, QC, Canada). After scanning, the fine roots were dried in an oven at 65 °C for a week, and the dry weight was measured. These values were used in calculating the following morphological trait values as a ratio between the acquired values from the image analysis and the dry weight: specific root length (SRL, a ratio between the total length and dry weight), specific root area (SRA, a ratio between the total surface area and dry weight), and root tissue density (RTD, a ratio between the total volume and dry weight). The remaining sampled fine roots that were collected by each distance were pooled in each treatment and used to evaluate the association of ECM. We classified the ECM taxa using the morphological characteristics that were observed with a microscope (Olympus szx-ILLK100, Tokyo, Japan). The relative abundance of ECM was calculated as the ratio of the number of ECM root tips to the non-ECM root tips. The ECM taxa were verified based on the methods of a previous study by Wang et al. [41].

## 2.4. Data Analysis

MS EXCEL 2010 (Microsoft ©) and R version 3.4.6 [49] were used for data processing and statistical analysis. Each response variable was calculated as the representative values per experimental block (five values for each treatment for statistical analysis). We evaluated the effects of the compaction and N loading by ANOVA with chi-squared test. ANOVA with chi-squared test was performed on SRL, SRA, and RTD between the horizontal distance from the root trunk and soil compaction at each N treatment. We analyzed Pearson's correlation coefficient between the three types of root density (total root density, lateral root density, and fine root density), SH of the first and final conditions, and HCI. The values in each N loading treatment were pooled for the correlation analysis at first in order to evaluate whether the relationships between the soil and root vary depending on compaction. A significant relationship was identified using the Bonferroni test. In the detected significant relationships, we analyzed whether the effects of the root traits on the soil properties vary by N loading using ANCOVA as a post-hoc test.

## 3. Results

### 3.1. Environment at the Study Site

Figure 1 shows the seasonal variations of the meteorological factor and surface SH. The total precipitation from May to November 2018 was 828 mm, and that from May to October 2019 was 482 mm. The compacted surface soil clearly showed a decline in hardness in 2018, whereas the variation was relatively small in 2019 (Figure 1). In 2018, the compaction treatments reduced the air fraction by more than 35% and increased bulk density by more than 50% (Table S1), whereas N loading increased

the total inorganic N contents by more than 20%. Therefore, the soil environment was successfully manipulated for the experimental design. Table S2 shows the variations of SH along with soil depth. A negative variation in SH in 2018 was found for high compaction plots: from 1.57 MPa for the surface soil layer to 0.51 MPa for the deepest layer. In 2019, the variation was positive at the control plots, and a negative variation was not observed at the high compaction plots.

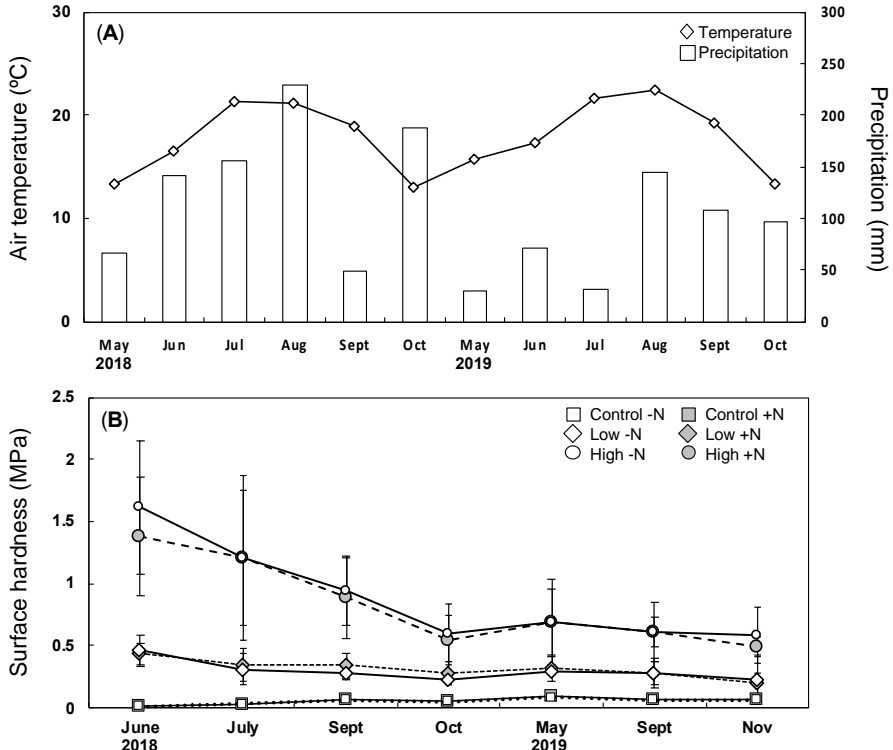

**Figure 1.** Meteorological and soil conditions at the study area. (**A**) The meteorological conditions in each month; Diamond: mean air temperature; Bar: total precipitation. (**B**) The seasonal changes in average surface SH ± standard error (SE); square: control; diamond: low compaction level; circle: high compaction level; white: non-N loading (−N); gray: N loading (+N). (*n* = 5).

*3.2. Aboveground Responses*

The effects of soil compaction and N loading were significant on most aboveground traits (Table 1). The final size of the seedlings was reduced in high compaction treatment by approximately 10.2% and 23.1% for the total height and collar diameter, respectively. Compaction significantly reduced the total dry mass by more than 30%. The interaction effects of compaction and N loading were observed ($p < 0.05$). The increased effects of N loading on dry mass were only significant at the control and not on compaction treatments. BPI was reduced by only the high compaction level under N loading, whereas NPI was significantly reduced by compaction alone ($p < 0.05$). A significant reduction of xylem water potential was observed in compacted soil at predawn (Table S3).

**Table 1.** Mean value ± SE of aboveground trait responses to soil compaction without N loading (−N) and with N loading (+N) and results of ANOVA (Compaction: C; N loading: N). BPI: branch productivity index; NPI: needle productivity index; ***: $p < 0.001$; **: $p < 0.01$; *: $p < 0.05$; N.S.: not significant. ($n = 5$).

| | | Control | | Low Compaction | | High Compaction | | ANOVA | | |
|---|---|---|---|---|---|---|---|---|---|---|
| **Aboveground Traits** | | **−N** | **+N** | **−N** | **+N** | **−N** | **+N** | **C** | **N** | **C × N** |
| Stem (cm) | Current | 157.59 ± 2.91 | 164.7 ± 6.08 | 160.09 ± 3.67 | 152.92 ± 5.97 | 151.25 ± 3.64 | 140.93 ± 8.05 | * | N.S. | N.S. |
| | 1-year | 42.64 ± 1.50 | 46.2 ± 2.13 | 43.94 ± 2.28 | 45.32 ± 2.39 | 35.91 ± 3.68 | 36.82 ± 4.54 | ** | N.S. | N.S. |
| | 2-year | 20.27 ± 0.75 | 20.53 ± 1.17 | 18.80 ± 2.18 | 20.14 ± 0.48 | 20.04 ± 0.74 | 20.29 ± 0.74 | N.S. | N.S. | N.S. |
| Diameter (mm) | Current | 16.47 ± 1.89 | 19.13 ± 2.10 | 16.96 ± 2.28 | 16.71 ± 2.35 | 15.90 ± 1.66 | 14.21 ± 2.82 | *** | N.S. | ** |
| | 1-year | 18.33 ± 6.20 | 22.97 ± 2.10 | 20.60 ± 2.09 | 20.10 ± 2.15 | 18.17 ± 2.91 | 16.68 ± 4.03 | ** | N.S. | ** |
| | Collar | 30.33 ± 2.85 | 34.45 ± 2.59 | 29.96 ± 3.44 | 29.75 ± 3.33 | 26.52 ± 3.07 | 23.34 ± 5.12 | *** | N.S. | ** |
| Total (g) | | 281.54 ± 16.89 | 386.49 ± 25.86 | 275.11 ± 14.70 | 277.53 ± 20.32 | 225.02 ± 17.04 | 213.15 ± 24.24 | *** | N.S. | * |
| Stem (g) | All | 166.56 ± 8.26 | 225.01 ± 14.25 | 171.18 ± 9.97 | 169.08 ± 10.48 | 135.86 ± 10.08 | 130.92 ± 12.14 | *** | N.S. | * |
| | Current | 56.27 ± 4.50 | 79.81 ± 6.79 | 61.56 ± 4.41 | 57.67 ± 4.46 | 50.51 ± 3.62 | 45.95 ± 4.36 | ** | N.S. | * |
| Branch (g) | All | 51.27 ± 5.11 | 73.91 ± 6.44 | 46.78 ± 2.81 | 51.16 ± 5.46 | 38.29 ± 3.19 | 33.45 ± 5.41 | *** | * | * |
| | Current | 13.26 ± 2.15 | 25.52 ± 4.34 | 15.14 ± 1.70 | 14.62 ± 2.92 | 17.61 ± 1.81 | 11.61 ± 2.41 | N.S. | N.S. | * |
| Needle (g) | All | 63.70 ± 5.54 | 87.56 ± 7.27 | 57.13 ± 3.63 | 57.28 ± 5.90 | 50.86 ± 4.90 | 48.77 ± 8.05 | *** | N.S. | N.S. |
| | Current | 50.53 ± 4.48 | 72.16 ± 6.52 | 47.11 ± 3.08 | 46.17 ± 4.94 | 43.61 ± 4.26 | 40.78 ± 6.43 | ** | N.S. | N.S. |
| Current stem | BPI ($m^{-1}$) | 16.18 ± 1.88 | 18.51 ± 1.68 | 16.54 ± 1.68 | 15.70 ± 1.61 | 17.82 ± 1.80 | 10.27 ± 1.32 | N.S. | N.S. | * |
| | Length (cm) | 67.05 ± 10.88 | 78.31 ± 18.05 | 92.24 ± 23.31 | 90.14 ± 26.95 | 69.31 ± 11.46 | 37.13 ± 3.22 | N.S. | N.S. | N.S. |
| Maximum branch | Diameter (mm) | 4.77 ± 0.32 | 5.01 ± 0.38 | 4.47 ± 0.20 | 4.26 ± 0.43 | 4.44 ± 0.26 | 3.64 ± 0.20 | N.S. | N.S. | N.S. |
| | Total (g) | 52.59 ± 4.64 | 74.28 ± 6.76 | 48.92 ± 2.96 | 47.89 ± 5.05 | 45.52 ± 4.51 | 41.72 ± 6.46 | ** | N.S. | N.S. |
| | Needle (g) | 2.06 ± 0.39 | 2.12 ± 0.44 | 1.80 ± 0.24 | 1.72 ± 0.31 | 1.91 ± 0.37 | 0.93 ± 0.08 | N.S. | N.S. | N.S. |
| | NPI ($g\ m^{-1}$) | 3.00 ± 0.20 | 3.11 ± 0.32 | 2.51 ± 0.17 | 2.36 ± 0.22 | 2.61 ± 0.17 | 2.50 ± 0.09 | * | N.S. | N.S. |

### 3.3. Belowground Responses

The effects of soil compaction were significant on all root densities (Table 2). Significant effects of N loading were not observed in lateral and fine root densities. Without N loading, the reduction of root density was approximately 50% in lateral and fine root, but less than 30% in the rootstock. The interaction effects of soil compaction and N loading were observed in both the rootstock ($p < 0.05$) and fine root density ($p < 0.001$). The effect of soil compaction was almost doubled under N loading in rootstock, but the fine root density under N loading was not changed by soil compaction. The lateral root proportion was significantly decreased by soil compaction ($p < 0.05$). The fine root proportion did not change significantly under soil compaction. However, the interaction effect of soil compaction and N loading was observed ($p < 0.05$). The reduction of fine root proportion due to soil compaction was not observed under N loading, rather its proportion was increased. The interaction effect of soil compaction and N loading was also observed in rootstock ($p < 0.05$), showing that N loading only increased its proportion without soil compaction.

**Table 2.** Mean value ± SE of underground responses to soil compaction without N loading (−N) and with N loading (+N) and the results of ANOVA (Compaction: C; N loading: N). ***: $p < 0.001$; *: $p < 0.05$; N.S.: not significant. ($n = 5$).

| Traits | Root Type | Control −N | Control +N | High Compaction −N | High Compaction +N | C | N | C × N |
|---|---|---|---|---|---|---|---|---|
| Root density (kg m$^{-3}$) | Total | 0.90 ± 0.08 | 1.16 ± 0.18 | 0.54 ± 0.07 | 0.68 ± 0.08 | *** | *** | N.S. |
| | Rootstock | 0.37 ± 0.06 | 0.60 ± 0.20 | 0.28 ± 0.09 | 0.33 ± 0.03 | *** | *** | * |
| | Lateral | 0.41 ± 0.07 | 0.46 ± 0.09 | 0.2 ± 0.02 | 0.26 ± 0.06 | *** | N.S. | N.S. |
| | Fine | 0.12 ± 0.02 | 0.09 ± 0.02 | 0.06 ± 0.01 | 0.08 ± 0.02 | *** | N.S. | *** |
| Root proportion to total root biomass (%) | Rootstock | 41.02 ± 6.27 | 51.9 ± 3.31 | 50.79 ± 4.78 | 49.32 ± 5.41 | N.S. | N.S. | * |
| | Lateral | 13.6 ± 2.82 | 8.47 ± 3.12 | 11.29 ± 2.63 | 12.27 ± 1.96 | * | N.S. | N.S. |
| | Fine | 45.38 ± 5.08 | 39.63 ± 3.03 | 37.92 ± 2.45 | 38.41 ± 4.34 | N.S. | N.S. | * |

### 3.4. Relationship between Soil Hardness and Roots

Correlation analysis showed several significant relationships between SH and root density (Table S4). The highest correlation was observed between the fine root proportion and surface HCI under compaction treatment (Figure 2B, $p < 0.001$). The same positive correlation was observed in the lateral root proportion under compaction treatment (Figure 2A, $p < 0.01$). The second highest correlation was between the 10–20 cm of HCI and total root density of the control ($p < 0.001$). The relationship between the 10–20 cm of HCI and fine root proportion was negative ($p < 0.001$). Similar tendencies of these relationships were not observed at the control or the compaction treatment (Table S4). The results of the post-hoc test showed that there was no significant effect of N loading on the relationship between HCI and each root proportion (Table S5).

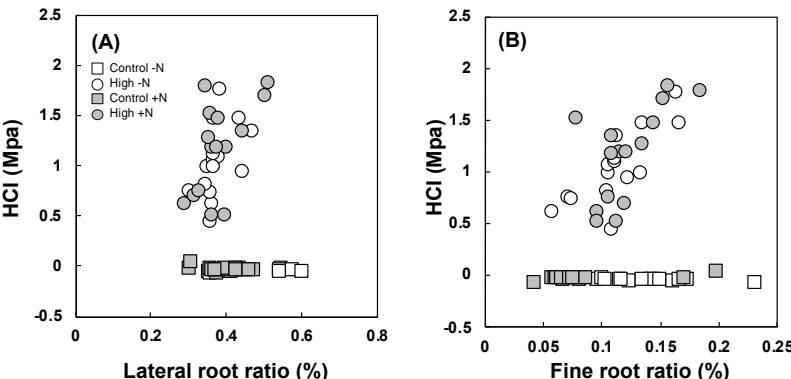

**Figure 2.** Relationships between (**A**) lateral root proportion and (**B**) fine root proportion and HCI ($n = 5$). Circle: control; square: high compaction; white: no N loading (−N); gray: N loading (+N).

### 3.5. Responses of Root Morphological Traits and ECM Association

SRL was significantly reduced along with the horizontal distance from the root trunk without N loading, regardless of soil compaction ($p < 0.05$, Table 3). The maximum reduction of SRL was approximately 40% (Figure 3A). By contrast, there were significant interaction effects of distance and compaction under N loading ($p < 0.001$), indicating that the root under N loading would develop depending on the SH. The maximum increment of SRL was nearly double under N loading (Figure 3B). The same pattern was observed in SRL, although there was no significant effect of distance without N loading. Moreover, RTD was significantly increased by soil compaction ($p < 0.05$, Table 3), where the average increment was approximately 20% (Figure 3E).

**Table 3.** Summary of ANOVA for responses of root morphological traits, specific root length (SRL), specific root area (SRA), and root tissue density. (RTD) without N loading (−N) and with N loading (+N). ***: $p < 0.001$; *: $p < 0.05$; N.S.: not significant. ($n = 5$).

| N Loading | Variable | SRL | | SRA | | RTD | |
|---|---|---|---|---|---|---|---|
| | | $\chi^2$ | *p* value | $\chi^2$ | *p* Value | $\chi^2$ | *p* Value |
| −N | Distance (D) | 4.23 | * | 1.79 | N.S. | 0.04 | N.S. |
| | Compaction (C) | 0.06 | N.S. | 0.51 | N.S. | 3.90 | * |
| | D × C | 0.27 | N.S. | 0.08 | N.S. | 0.03 | N.S. |
| +N | D | 0.27 | N.S. | 0.67 | N.S. | 0.85 | N.S. |
| | C | 1.92 | N.S. | 0.38 | N.S. | 0.29 | N.S. |
| | D × C | 11.75 | *** | 5.18 | * | 0.74 | N.S. |

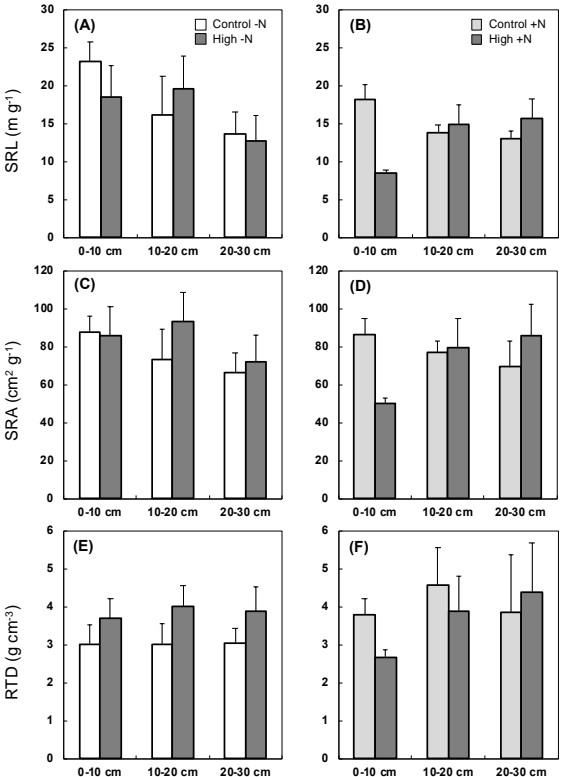

**Figure 3.** Mean value ± SE in specific root length (SRL), specific root area (SRA), and root tissue density (RTD) ($n = 5$). The responses were showed without N loading (**A,C,E**) and with N loading (**B,D,F**). White and light gray: no compaction; dark gray: high compaction.

The ECM taxa were composed of 55 organisms that were identified to species levels, 23 identified to genus level, and 18 as anonymous fungi (Table S6). The most abundant species was *Wilcoxina*,

except for the soil compaction and N loading sites (Table 4). The genus *Suilus*, a larch-specific ECM, was the second most abundant, especially at soil compaction and N loading sites.

**Table 4.** Results of relative abundance (%) for top five in associated ECM under soil compaction and N loading (*n* = 5). The identified fungal taxa were described as genus level.

| Control | | | | High Compaction | | | |
|---|---|---|---|---|---|---|---|
| **−N** | | **+N** | | **−N** | | **+N** | |
| *Wilcoxina* | 33.87 | *Wilcoxina* | 31.42 | *Wilcoxina* | 34.07 | *Suillus* | 42.73 |
| *Suillus* | 23.44 | *Suillus* | 19.14 | *Suillus* | 16.31 | *Wilcoxina* | 15.86 |
| *Laccaria* | 10.10 | *Dothideomycetes* | 17.06 | *Inocybe* | 11.82 | *Tetracladium* | 10.77 |
| *Helicorhoidion* | 5.41 | *Epicoccum* | 8.81 | *Tomentella* | 8.34 | *Saitozyma* | 6.32 |
| *Articulospora* | 3.46 | *Tomentella* | 5.14 | *Tricholoma* | 7.08 | *Cryptococcus* | 3.45 |

## 4. Discussion

### 4.1. Relationship between Roots and Soil

Significant effects of root development on soil physical parameters were found under soil compaction treatments (Figure 2, Table S4). Namely, the root developments of $F_1$ seedlings could enhance the recovery of compacted soil, which was supported by previous studies [28–30]. However, in contrast to the results that were found with alder species [30], almost no effects were found in soil condition at a deeper layer (Table S4). Our results would be associated with the shallow root system of larch [34]. Dahurian larch, as one parent of $F_1$, would develop a shallow root system as an adaptation to a thinner soil layer in the presence of permafrost [34,35]. In fact, there were hardly any roots deeper than the excavating depth of 40 cm when root systems were dug up. Rather, the root systems under soil compaction stress showed the shrunken shape. These results indicated that the effects of $F_1$ root developments on soil recovery would mainly occur in shallow soil layers. Further, the effects of N loading were not significant on the relationship between HCI and root traits (Table S5), which could be supported by the persistent relationship between roots and soil physical parameters under N application [50]. Soil recovery induced by root development of planted $F_1$ might not be associated with soil N condition, even though N loading increased lateral and fine root proportions [42,43]. Possible reasons for the persistent relationship between root development and soil recovery could be linked to more specific root morphological responses to soil compaction and N loading.

Relatively shallow soil layers could regenerate faster than the deeper layer [51] because of the soil surface-specific events, such as swelling, shrinking, and freezing [1,6,20]. These studies supported our results that the seasonal physical changes of surface soil were significant (Figure 1). Indeed, the significant physical changes of surface soil in the first year may be partly due to the relatively high amount of precipitation [1,8,20]. However, there were non-significant effects of N loading on most physical soil parameters (Tables S1 and S2). These results would be supported by previous studies [51]. The effects of chemical treatment on forest soil have been mainly evaluated from the lime application [52,53], and little is known regarding N application [31,32]. Further, soil physical dynamics could be affected by biological and chemical factors, like soil macro- and micro-fauna [7,28]. Future studies are needed in order to elucidate the effects of N addition on the physical properties of forest soil, focusing on the interaction between soil chemical and biological factors [22].

### 4.2. Root Morphological Developments and ECM Symbiosis

It has been reported that there are two types of root development under soil compaction [29]: (i) an avoidance type with higher root developments for allowing the resource exploration in compacted soils and (ii) a resistance type with lower root morphological traits for improving root growth per acquired resources, such as water and nutrients. Non-significant changes in SRL and SRA without N loading (Figure 3) indicated that the studied $F_1$ seedlings might be a resistance type that is associated

with lower plasticity in root development [29]. However, contrasting responses occurred with N loading. That is, root development under soil compaction would vary depending on N loading, which means that the physical response capacity of the root would be altered by soil chemical factors. The morphological response at a specific root level can be interpreted as a strategy for efficient resource acquisition [29,54]. We assumed that $F_1$ seedlings would acclimate to soil compaction via both the carbon allocation changes within the root system and specific root morphological changes, resulting in a persistent relationship between root development and soil recovery. Moreover, the diversity of ECM-associated $F_1$ seedlings under soil compaction was changed by N loading (Table 4). The ECM community may vary between different soil nutritional conditions, i.e., N and P [40]. In particular, the specific symbiotic relationship with *Suillus* sp. was reported in larch species [41], which may accelerate the root development of host plants [55]. The shorter branching fine roots of $F_1$ seedlings may be associated with the specific ECM community under soil compaction and N loading given that ECM fungi could induce a higher frequency of root tips. Because the effects of association and treatments could not be separated in this study, further investigations are required to more accurately understand the root–ECM interaction. In the future, the quantitative effects of root morphological traits on soil recovery also need to be evaluated [1,24,56].

### 4.3. Aboveground Responses

Different response patterns between NPI and BPI without the N loading condition indicated that BPI would be persistent more than NPI under compacted soil. Branch number is mostly determined by the number of buds and its flush condition [57]. Thus, the persistent BPI indicated that soil compaction would not suppress shoot branching. It has been reported that soil compaction stimulated the regulation of aboveground growth via phytohormonal [4,58]. The plastic responses of shoot branches to SH may be a cue for investigating the interaction between above- and below-ground growth with the complex hormonal regulations. The lower NPI under soil compaction indicated that soil compaction would inhibit the leaf production and growth in $F_1$ via water stress. Given the lower xylem water potential (Table S3), the photosynthesis of $F_1$ seedlings would be suppressed by soil compaction, resulting in NPI reduction.

### 5. Conclusions

Significant positive correlations between both fine and lateral root proportion and HCI indicated that the root development of $F_1$ seedlings was associated with the soil recovery after compaction. N loading promoted root development, especially under soil compaction. N loading increased fine root density and its proportion under soil compaction, which might be associated with high susceptibility to drought [43]. Indeed, a significant interaction effect of soil compaction and N loading was observed on current-year growth parameters (Table 1). The effects of lower precipitation and soil moisture condition during the growing period in 2019 were also significant (Figure 1). As the water condition was not manipulated in this study, future studies are needed in order to verify the responses to drought tolerance under soil compaction and N loading, where the relationship between above-, and belowground responses should be linked. N loading also changed specific root morphological traits, which could interact with ECM symbiosis, resulting in the retention of the relationship between SH and root traits, even with high proportions of fine roots. Our study addressed the understanding of the capacity of planted $F_1$ seedlings for soil recovery in ecological silviculture management.

**Supplementary Materials:** The following are available online at http://www.mdpi.com/1999-4907/11/9/947/s1, Figure S1: Experimental design of the study site. Table S1: The soil physical and chemical properties in 2018 (*n* = 5). Table S2: The SH and HCI along with soil depth (*n* = 5). Table S3: Responses of xylem water potential in 2019 (*n* = 5). Table S4: Results of the correlation analysis between root traits and SH. Table S5: Results of the post-hoc test evaluating the effect of N loading on the relationship between HCI and root proportion under compaction (*n* = 5). Table S6: The lists of all identified ECM fungal taxon.

**Author Contributions:** T.S. and T.K. conceived the experiment. T.S., S.Y., Y.T., H.M., and T.K. conducted the experiments, measurements, and data analysis. F.S. managed the experimental site. T.S. wrote the first draft of the

paper. Y.T., E.M., T.W., and T.K. jointly revised and edited the paper. All authors have read and agreed to the published version of the manuscript.

**Funding:** This work was partly funded by the Japan Forest Technology Association, Japan Society for the Promotion of Science, KAKENHI (Proj. No. 18J2013908), and Strategic International Collaborative Research Program of the Japan Science and Technology Agency (Grant No. JPMJSC18HB).

**Acknowledgments:** We thank K. Ichikawa, E. Fujito and the staff of the Sapporo Experimental Forest of Hokkaido University for the management of nursery. Technical supports from T. Nakaji and E. Agathokleous, and helpful discussion and comments from H. Maruyama, M. Ohashi, T. Yoshida, T. Shinano and the special colleague S. Fujita and R. Doi are gratefully acknowledged. Thanks are also due to HRO Forestry research institute kindly offered seedling samples as a part of their breeding effort as well as E. Paoletti and Y. Hoshika of Institute of Research on Terrestrial Ecosystems of CNR for her kind arrangement of the international cooperation.

**Conflicts of Interest:** The authors declare no conflict of interest.

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
