# Peer review of "Evaluating Soil–Root Interaction of Hybrid Larch Seedlings Planted under Soil Compaction and Nitrogen Loading"

_forests, doi:10.3390/f11090947_

Round 1

Reviewer 1 Report

The manuscript follows the effects of soil compaction and N load on the growth of larch seedlings and their root system morphology. Vice versa it analyses the effect of root growth on soil recovery from the compaction. The study presents interesting data from two-year experimental period. It is clear that authors worked really hard on the experimental site. The manuscript however needs to be improved before publication to increase the comprehensibility of the text. And it desires the corrections by a native English speaker. In summary, I recommend to accept manuscript after revision of the text. Detailed comments are below.

Comments to the text:

Introduction is rather long. Should be shortened to avoid repetition of information. It should go more to the point.

Methods are described thoroughly, sometimes including too many details. Make them shorter and more comprehensive.  I would suggest to mark the treatments without N addition as –N instead of +W (in the whole manuscript). +W is confusing as water was added in all treatments.

Results:

  1. The number of repetitions should be mentioned in each figure/table information.
  2. The description of the treatments in the comment to the graph is useless if there is a legend inside the graph. Avoid the repetition of information.
  3. Specify what temperature fig.1 presents.
  4. Why root section do not show data for root biomass fractions in a similar arrangement as for aboveground biomass data. It is difficult to see changes in root density, lateral root proportion etc. in response to treatments without seeing the numbers. Relationships between parameters are interesting, but I would like to see the means for individual treatments as well (perhaps in similar arrangement as in table S3).
  5. Table3 is unclear and perhaps useless. Are there any bold letters (indicating the significance)? I cannot see any. If not, I would skip this table and comment the trends in the text. “HCI:” at the end of the table legend – something seems missing. Are last two rows of this table correct? There are same numbers as in first two rows.
  6. Table S2 presents many similar SE. Is it correct?
  7. Table S1 and S2 miss statistical analyses.

Discussion is difficult to read. You have interesting data and you can make the output of your study clearer. It seems that soil compaction was the limiting factor determining root growth more strongly than nitrogen addition did, but it is difficult to see from the presented graphs and tables as well as from the discussion and conclusions. What does mean HIS (line 404)?

Author Response

Thank you very much for your constructive suggestion. We deeply appreciating reviewers as well as your comments and suggestions. We totally agreed with your suggestions and revised as much as possible.

The responses to comments listed below reflect the content of the revised new manuscript. We hope you will approve the upload again.

Comments: Introduction is rather long. Should be shortened to avoid repetition of information. It should go more to the point.

Response: We reconstructed the introduction, material and Methods, Discussion, and Conclusion by eliminating repetition of information.

Comments: Methods are described thoroughly, sometimes including too many details. Make them shorter and more comprehensive.

Response: We modified the method by reducing its volume, especially a detail explanation of ECM gene analysis, and have added the supplemental figure although the part of soil environment analysis has been described for clearer with a minimum amount based on the suggestion of the reviewer 2.

Comments: I would suggest to mark the treatments without N addition as –N instead of +W (in the whole manuscript). +W is confusing as water was added in all treatments.

Response: We have modified from +W to –N for the treatment without N addition. [Line 109]

Comments: The number of repetitions should be mentioned in each figure/table information. The description of the treatments in the comment to the graph is useless if there is a legend inside the graph. Avoid the repetition of information.

Response: We added the number of repetitions (i.e. number of treatment replication) in each figure and table.

Comments: Specify what temperature fig.1 presents.

Response: We added the specific title for figure 1 as “air temperature”

Comments: Why root section do not show data for root biomass fractions in a similar arrangement as for aboveground biomass data. It is difficult to see changes in root density, lateral root proportion etc. in response to treatments without seeing the numbers. Relationships between parameters are interesting, but I would like to see the means for individual treatments as well (perhaps in similar arrangement as in table S3).

Response: We separated part 3.3 for a part for the results of root responses [Line 254-271] and another part for relationships between root trait and soil properties. [Line 272-285]

Comments: Table3 is unclear and perhaps useless. Are there any bold letters (indicating the significance)? I cannot see any. If not, I would skip this table and comment the trends in the text. “HCI:” at the end of the table legend – something seems missing. Are last two rows of this table correct? There are same numbers as in first two rows.

Response: Thank you for your comments and sorry for confusion from the miss-edited Table. We transferred Table 3 to supplemental data (Table S4) although there were several significant relationships between root trait and soil properties. Rather, we have shown the most significant and clarify relationships between root trait and HCI in Figure 2. Results of post-hoc test have also been transferred to supplemental data (Table S5).

Comments: Table S2 presents many similar SE. Is it correct?

Response: As some unit conversions were calculated wrong, so we improved them.

Comments: Table S1 and S2 miss statistical analyses.

Response: We added the results of statistical analyses.

Comments: Discussion is difficult to read. You have interesting data and you can make the output of your study clearer. It seems that soil compaction was the limiting factor determining root growth more strongly than nitrogen addition did, but it is difficult to see from the presented graphs and tables as well as from the discussion and conclusions.

Response: As indicated, we reconstructed the Discussion parts, especially in the first paragraph [Line 315-342], where the two paragraphs were established for the relationship between root and soil parameters and the response of soil properties to treatments.

Comments: What does mean HIS (line 404)?

Response: Sorry miss-typing. We modified it to HCI. [Line 326]

Reviewer 2 Report

The article by Sugai et al deal with the use of larix seedlings planted under different soil compaction densities and Nitrogen addition to evaluate their use for recovering compacted soil. The article is quite well structured and of interests for the readers of FORESTS.

The introduction is quite well focused on the effect that soil compaction can have on soil physical properties, considering the use of different species such as N fixing species and/or the application of fertilizer. Some of the sentences are not very clear (see specific comments).

In the material and method section some parts needs to be better explained. It is not very clear to me where the N addition was performed. In which compacted soil treatment it was performed? Concerning the soil sampling a more clear explanation has to be provided for some basic soil parameters that are at the core of the study. For instance is not very clear how bulk density was measured. Also the soil properties determinations are not very clear. Why soil samples were not sieved at 2 mm as for standard analyses? The BPI and NPI indexes were developed in this study or are currently used and developed by other studies?

The discussion is quite well developed. The main concern is related to the first paragraph of the discussion were it is described the relationship between root and soil parameters. An explanation should be given for the comparison of the same depths under different degree of soil compaction. After the soil is compacted, the authors compare depths, which comprise a different volume of soil due to the compaction. Some effects (e.g effect of nitrogen addition) could be related to the fact that there is more soil in the same considered depth. Some clarifications should be made.

The article need to be revised by a native speaker since some technical terms are not correct and many sentences throughout the article not clear at all.

Line 45-46: which are the 17 woody plant species? This sentence in the present form does not fit here. Rephrase or remove

Line 62-63: I assume that the choice of the species (alnus sp) to be adopted can vary depending on the location.

Line 113-114: Soil were classified. Not Soil profile since you did not introduced that profiles were digged. Based on for soil resource?

Line 146: what means +W and +N?

Line 147: What is HN?

Line 167-175: It is not very clear how the soil hardness was measured with depth. Minit pits were excavated and the SH measured horizontally? Why an acrylic box was then introduced in the pit excavated? What is the purpose? This part should better explained

Line 183-188: to measure BD a cylinder of known volume was used. But then at line 188 it is stated that a digital volume analyser was used to measure the solid phase. This part is a bit confusing. Please, make clear how these parameters where measured and which instruments/methodologies used.

Line 189-195: To measure soil properties, some samples were collected. Why the samples where not treated as it is usually done for all the soil analyses (e.g. sieving at 2 mm)? Where the soil stone free? This part should be better developed since the soil is at the core of the article.

Line 219: Based on the significant differences of above-ground biomass…. Where these differences were showed. No any difference was introduced at the moment concerning above-ground biomass

Line 290: meteorological factors

Line 294-295: could the N increase be related to soil compaction? If the compaction increases, then the final soil volume for a specific depth you compare is not the same but is much more due to compaction.

Line 390: dug out?

Line 403-405: please rephrase this sentence

Author Response

Thank you so much for your constructive comments. We would like to appreciate all reviewer’s suggestions and insightful comments. Based on these comments, we have revised as much as possible.

*The responses to comments listed below reflect the content of the revised new manuscript. We hope you will approve the upload again. 

Comments: Line 45-46: which are the 17 woody plant species? This sentence in the present form does not fit here. Rephrase or remove

Response: We deleted this sentence.

Comments: Line 62-63: I assume that the choice of the species (alnus sp) to be adopted can vary depending on the location.

Response: The species selection for planting seedlings varies depending on its purpose as well as the location. In previous studies, however, the major target for evaluating capacities of soil recovery has been occupied by Alder (Alnus sp.) or other broadleaves (e.g. Meyer et al. 2014 Soil and Tillage Research, Fernandez et al. 2019 Forest Ecology and Management, Walro et al. 2019 Forests, Jourgholami et al. 2020 New Forests) whereas little is known in conifer seedlings for artificial forests regarding the potential for recovering compacted soil. Thereby, we focused on the response of hybrid larch seedling.

Comments: It is not very clear to me where the N addition was performed. In which compacted soil treatment it was performed?

Response: The point of N addition was set under non-compaction and compaction treatments. As you indicated, we reconstructed the part of material and method with new supplemental figures (Figure S1) for visually understanding of study design.

Comments: Concerning the soil sampling a clearer explanation has to be provided for some basic soil parameters that are at the core of the study. For instance, is not very clear how bulk density was measured. Also, the soil properties determinations are not very clear.

Response: As indicated, we improved the detail methods for each soil analysis [Line 116-153].

Comments: Why soil samples were not sieved at 2 mm as for standard analyses?

Response: When we measured soil physical properties, we tried to preserve the original form of soil core sample. However, when we measured soil chemical properties (i.e. pH and soil inorganic N contents), we have conducted the soil sieving by a stainless-steel sieve with 1.0 mm mesh [Line 147-153].

Comments: The BPI and NPI indexes were developed in this study or are currently used and developed by other studies?

Response: The idea for BPI and NPI was developed in this study, and in our knowledge, the same index was not provided in other studies of soil compaction.

Comments: Line 113-114: Soil were classified. Not Soil profile since you did not introduce that profiles were dogged. Based on for soil resource?

Response: Sorry for miss-typing. We classified the soil of experimental site based on World reference base for soil resources. [Line 81-82]

Comments: Line 146: what means +W and +N?

Response: We revised the abbreviation from +W to –N, indicating the treatment without N addition. [Line 109]

Comments: Line 147: What is HN?

Response: Sorry for miss-typing. We have modified [Line 111].

Comments: Line 167-175: It is not very clear how the soil hardness was measured with depth. Minit pits were excavated and the SH measured horizontally? Why an acrylic box was then introduced in the pit excavated? What is the purpose? This part should be better explained

Response: We have firstly removed the part of an acrylic box for more clarify understanding. As indicated, we have reconstructed more clarify contents regarding the measurement of vertical soil hardness [Line 121-133] as following;

“The vertical SH was measured along with four depths: 0–10, 10–20, 20–30, and 30–40 cm. These measurements were taken during the first and final research periods. The first SH along with depth were measured on May 15, 2018. The hole space was manually dug into soil with a volume of approximately 10 cm × 35 cm × 45 cm. The position of the hole was randomly set in each subplot at least 30 cm away from the planted seedling. Then, SH was randomly measured in a soil face at each depth. The mean value of these measurements was used as a representative SH value for each seedling. The final SH along with depth was measured in a space with a volume of approximately 50 cm × 50 cm × 40 cm from the end of October to early November in 2019, where a seedling was excavated. The three points where the SH was measured were selected randomly at each depth after excavating a seedling, and the mean value was calculated for each seedling. To evaluate the relationship between soil variation and root development considering soil depth, the hardness change index (hereafter HCI) was calculated as the difference between the final and first mean values of SH.”

Comment: Line 183-188: to measure BD a cylinder of known volume was used. But then at line 188 it is stated that a digital volume analyzer was used to measure the solid phase. This part is a bit confusing. Please, make clear how these parameters where measured and which instruments/methodologies used.

Response: We separated paragraphs for each soil analysis (i.e. soil bulk density and soil water content, soil fraction, and soil chemical properties) [Line 138-153].

Comment: Line 189-195: To measure soil properties, some samples were collected. Why the samples where not treated as it is usually done for all the soil analyses (e.g. sieving at 2 mm)? Where the soil stone free? This part should be better developed since the soil is at the core of the article.

Response: When we measured soil chemical properties (i.e. pH and soil inorganic N contents), we have conducted the soil sieving by a stainless-steel sieve with 1.0 mm mesh. On the other hands, we tried to preserve the original form of soil core sample avoiding the changes in physical properties induced by sampling disturbance. Since we have understood the significant effects of large particles (e.g. stone) on soil physical properties, we have conducted a root tillage treatment for eliminating these noise effects as much as possible. Further, when we measured soil physical parameters, we hardly observed the obvious large particles in any samples.

Comment: Line 219: Based on the significant differences of above-ground biomass…. Where these differences were showed. No any difference was introduced at the moment concerning above-ground biomass

Response: We modified them as following:

“The root systems of the seedlings were carefully excavated at the control and at a high level of soil compaction treatment from the end of October to early November in 2019.” [Line 177-178]

Comment: Line 290: meteorological factors

Response: We changed it. [Line 223]

Comment: Line 294-295: could the N increase be related to soil compaction? If the compaction increases, then the final soil volume for a specific depth you compare is not the same but is much more due to compaction.

Response: Based on the new statistical results of Table S1 and S2, most soil physical parameters were not changed by N loading in this study. However, the viewpoint whether the physical parameters of forest soil would be affected by N loading would be a key for more deeply understanding of soil recovery, we think. We described detail below.

Comment: The main concern is related to the first paragraph of the discussion were it is described the relationship between root and soil parameters. An explanation should be given for the comparison of the same depths under different degree of soil compaction. After the soil is compacted, the authors compare depths, which comprise a different volume of soil due to the compaction. Some effects (e.g effect of nitrogen addition) could be related to the fact that there is more soil in the same considered depth. Some clarifications should be made.

Response: Thank you very much for your insightful comments. We firstly separated the first paragraph into two parts regarding the relationship between root and soil parameters and the response of soil properties to treatments. In fact, the dynamics of soil properties itself could be contributed to deeply understandings of soil recovery after soil compaction. In our results, relationships between soil chemical and physical parameter would not be clear, indicating that N addition would not have significant influence on soil physical properties. While, soil physical dynamics would be affected by biological and chemical factors as like soil macro-, micro fauna. Thereby, we consider that further studies are needed to elucidate the effects of N addition on the physical properties of forest soil, focusing on the interaction between soil chemical and biological factors. [Line 332-342]

Line 390: dug out?

Response: Thank you. We modified [Line 323].

Line 403-405: please rephrase this sentence

Response: We modified to HCI [Line 326].

Round 2

Reviewer 2 Report

Thanks for submitting the revised version of this manuscript.

The Introduction is now improved fixing the small flows of the previous version.

The materials and methods section is improved too. A clear explanation of how N addition and the soil parameters were determined is now provided. The BPI and NPI indexes are now clearer and better presented.

The Discussion, particularly for the first section, sounds much better now. The improvements performed by the authors notably increased the quality of this section.

Specific comments

Line 37: “contrasting” rather than “contrast”

Line 44: change “soil texture and organic matter content” with “soil texture, organic matter content,”

Line 252: change “under to soil compaction” with “under soil compaction”

Line 254: change “we should investigate the relationship between” with “it is of outmost importance to investigate the relationship between”

Line 282: “The soil of the experimental area” rather than “the soil profile…..”

Line 651: “A mini pit was manually dug” rather than “The hole space was manually dug”

Figure 1: remove the black background

Figure 3: remove the black background. It is not possible to see the SE bars

Line 1949: “These studies…” rather than “These reports…”

Author Response

Dear Reviewer2
We would appreciate all your helpful comments. We have revised all the specific points as follow.
Again, thank you so much for your kindly review.
Tetsuto SUGAI and co-authors
Corresponding author: Research Faculty of Agriculture, Hokkaido University, Kita 9 Nishi 9, Sapporo,
Hokkaido, 060-8589, Japan, E-mail: [email protected]
==================
Response to specific comments (SC)
#SC1-Line 37: “contrasting” rather than “contrast”
Thank you. We have changed it on new line 27.
#SC2-Line 44: change “soil texture and organic matter content” with “soil texture, organic matter
content,”
We have modified it on new line 43.
#SC3-Line 252: change “under to soil compaction” with “under soil compaction”
Sorry for our miss-typing. We have deleted “to” on new line 51.
#SC4-Line 254: change “we should investigate the relationship between” with “it is of outmost
importance to investigate the relationship between”
We have changed it on new line 52-53.
#SC5-Line 282: “The soil of the experimental area” rather than “the soil profile…..”
We have modified it on new line 81.
#SC6-Line 651: “A mini pit was manually dug” rather than “The hole space was manually dug”
We have modified it on new line 122.
#SC7-Figure 1: remove the black background
Thank you for your suggestion. We have changed the color from black to white and modified the
caption for showing the air temperature as a diamond symbol.
#SC8-Figure 3: remove the black background. It is not possible to see the SE bars

Thank you. We have changed the color from black to dark gray for showing the treatment of soil
compaction.
#SC9-Line 1949: “These studies…” rather than “These reports…”
We have modified it on new line 332
